# Determinants of left ventricular hypertrophy in a cohort attending a medical clinic: A comparative analysis stratified by sex

Sydney Mulamfu[1,2]◉, David Chisompola◉[1,2]*◉, Martin Chakulya[1,2], John Nzobokela[1,2], Phinnoty Mwansa◉[1,2], Benson M. Hamooya[1,2,3], Joreen P. Povia[4], Sepiso K. Masenga[1,2]*

1 Department of Cardiovascular Science and Metabolic Diseases, Livingstone Center for Prevention and Translational Science, Livingstone, Zambia, 2 HAND Research Group, School of Medicine and Health Sciences, Mulungushi University, Livingstone, Zambia, 3 Department of Public Health and Biostatistics, Livingstone Center for Prevention and Translational Science, Livingstone, Zambia, 4 Department of Health Economics, Livingstone Center for Prevention and Translational Science, Livingstone, Zambia

◉ These authors contributed equally to this work.
* d.chisompola@gmail.com (DC); sepisomasenga@gmail.com (SKM)

## Abstract

Left ventricular hypertrophy (LVH) is a significant cardiovascular complication in sub-Saharan Africa. However, sex-specific determinants of LVH in this population remain poorly understood. This study aimed to identify factors associated with LVH in a cohort attending medical clinic, stratified by sex. We conducted a cross-sectional analysis of 333 adults attending an outpatient clinic at Livingstone Teaching Hospital, Zambia. LVH was diagnosed by echocardiography. Data on demographics, clinical history (HIV, TB, hypertension) and metabolic profiles were collected. Univariable and multivariable logistic regression models were used to identify predictors of LVH in the overall cohort and in sex-stratified subgroups. $p < 0.05$ was considered statistically significant. Data from 333 participants were analysed. LVH was diagnosed in 75 participants (22.5%). In the overall cohort, independent factors associated with LVH were older age (AOR: 1.06 per year, 95% CI: 1.03–1.09, $p < 0.0001$), female sex (AOR: 2.31, 95% CI: 1.13–4.68, $p = 0.020$), hypertension (AOR: 2.52, 95% CI: 1.36–4.66, $p = 0.003$), and heart failure (AOR: 7.50, 95% CI: 1.85–30.3, $p = 0.005$). Sex-stratified analysis revealed distinct profiles: in males, LVH was significantly associated with older age (AOR: 1.06, 95% CI: 1.00–1.13, $p = 0.030$), hypertension (AOR: 3.70, 95% CI: 1.03–13.2, $p = 0.044$), and heart failure (AOR: 18.3, 95% CI: 1.60–209.0, $p = 0.019$), whereas in females, factors associated with LVH included older age (AOR: 1.06, 95% CI: 1.03–1.09, $p < 0.0001$), hypertension (AOR: 2.50, 95% CI: 1.21–5.18, $p = 0.013$), waist circumference (AOR: 1.05 per cm, 95% CI: 1.00–1.10, $p = 0.047$), and elevated cholesterol-HDL ratio (AOR: 1.53, 95% CI: 1.02–2.30, $p = 0.036$). LVH in this cohort was associated with age, female sex, hypertension, and heart failure. Factors associated with LVH differed by sex, with metabolic factors playing a more prominent role in women and hemodynamic factors in men. These findings underscore the

**Data availability statement:** No - some restrictions will apply; The datasets generated and/or analyzed during the current study are not publicly available due to ethical and privacy restrictions related to the use of human participant data, as governed by the Institutional Review Board (IRB) of Mulungushi University. The consent provided by participants and the approved study protocol do not permit unrestricted public data sharing. However, de-identified data may be made available to qualified researchers upon reasonable request, subject to review and approval by the Mulungushi University School of Medicine and Health Sciences Research Ethics Committee, and the completion of a formal data sharing agreement to ensure compliance with ethical and confidentiality requirements. Requests for access to the data can be directed to: Mulungushi University School of Medicine and Health Sciences Research Ethics Committee, Akapelwa Street, Livingstone, Zambia 10101; Phone: +260 967758554; Email: mmiyoba@mu.ac.zm.

**Funding:** This work was supported by National Institute of Diabetes and Digestive and Kidney Diseases of the National Institutes of Health grants (R21TW012635 to SKM), and the American Heart Association Award Number (24IVPHA1297559 to SKM). The funders had no role in study design, data collection and analysis, decision to publish, or preparation of the manuscript.

**Competing interests:** The authors have declared that no competing interests exist.

importance of sex-specific cardiovascular risk assessment and tailored management strategies.

## Introduction

Left ventricular hypertrophy (LVH) is a well-established marker of subclinical cardiovascular disease and an independent predictor of adverse cardiac outcomes, including heart failure, arrhythmias, and sudden cardiac death [1,2]. It is estimated to affect 10–20% of adults, with rates significantly higher among individuals with hypertension. Approximately 80% of cases occur in low- and middle-income countries (LMICs) [3,4].

In the general population, LVH is primarily driven by chronic pressure overload, often secondary to hypertension, and is influenced by traditional cardiometabolic risk factors such as obesity, dyslipidaemia, and diabetes mellitus [5]. Among people living with HIV (PLWH), the prevalence and pathophysiology of LVH appear to be more complex. While antiretroviral therapy (ART) has dramatically improved life expectancy, emerging evidence suggests that PLWH face an elevated risk of cardiovascular diseases, including LVH [6]. This elevated risk may be attributed to a confluence of factors, including traditional cardiovascular risk factors, direct and indirect effects of chronic HIV infection, ART-related metabolic alterations, and chronic systemic inflammation. However, the relative contributions of these factors, and whether they differ by sex, remain incompletely understood.

Sex differences in cardiovascular physiology and disease presentation are increasingly recognized [6]. Women and men differ in patterns of cardiac remodelling, risk factor profiles, and clinical outcomes [7,8]. In HIV, sex-specific differences in body composition, immune activation, and ART pharmacokinetics may further modulate cardiovascular risk. Despite this, most studies on HIV-associated cardiac complications have either not stratified by sex or have been underpowered to detect sex-specific determinants of LVH.

Thus, there is a need for studies that systematically evaluate the determinants of LVH in clinic-attending populations with explicit consideration of sex differences. Such insights could inform more personalized risk stratification and targeted interventions. Therefore, we aimed to identify the clinical, metabolic, and HIV-related factors associated with LVH in a cohort of individuals with and without HIV, and to perform a comparative analysis stratified by sex. We hypothesized that the determinants of LVH would differ meaningfully between males and females, with implications for sex-specific cardiovascular risk management in this population.

## Materials and methods

### Study design and setting

This was a hospital-based cross-sectional study conducted at the outpatient medical clinic of Livingstone University Teaching Hospital from October 1, 2023, to June 1, 2024. A total of 333 adults attending routine medical check-ups who provided

written informed consent were enrolled. The study was designed and reported in accordance with the Strengthening the Reporting of Observational Studies in Epidemiology (STROBE) guidelines for cross-sectional studies supplementary file (S1 File).

### Study participants and eligibility criteria

Participants were included if they were 18 years or older and had given their written informed consent. The study group consisted of PLWH and HIV-negative individuals (PWTH), regardless of hypertension status. PLWH needed to have been on stable antiretroviral therapy for at least 6 months and to have an undetectable viral load (<50 copies/mL). PWTH status was verified through standard tests. Individuals were excluded if they had diabetes, were pregnant, had severe kidney or liver disease, active opportunistic infections, or any other major health conditions that might affect systemic inflammation or cardiovascular measures.

### Sample size and sampling

A purposive (non-probability) sampling technique was employed to facilitate efficient enrollment. The sample size was calculated using the single population proportion formula. An expected prevalence of 23% [9] was assumed based on prior evidence. A 95% confidence level (Z = 1.96) and a margin of error of 5% were used. The minimum required sample size was calculated as follows:

$$n = \frac{Z^2 X\, p(1-p)}{d^2}$$

Where n represents the required sample size, p the estimated prevalence, Z the standard normal deviate corresponding to a 95% confidence level, and d the margin of error. Substitution of these values yielded a minimum sample size of 272 participants. To account for possible non-response or incomplete records, a 10% contingency was added, resulting in a final target sample size of 300 participants. To ensure robustness for subgroup analyses and account for potential data issues, we recruited 333 participants, providing ample statistical power.

### Hypertension definition

Hypertension was defined as systolic blood pressure ≥140 mmHg and/or diastolic blood pressure ≥90 mmHg based on the average of three seated measurements taken five minutes apart [10], or current use of antihypertensive medication(s) as documented in medical records or self-reported during the study interview.

### Biochemical and biomarker assays

Venous blood specimens were collected, aliquoted and stored at −80°C until analysis. Inflammatory cytokines such as IL-17A, IL-6 and components of the renin-angiotensin-aldosterone system (renin, angiotensin II, aldosterone) were measured using commercial enzyme-linked immunosorbent assay (ELISA) kits Elabscience (Elabscience Biotechnology Inc., China), according to manufacturer instructions. Lipid profiles such as total cholesterol triglycerides, and Very-low-density lipoprotein (VLDL) were analyzed with standard automated clinical chemistry platforms.

### Left ventricular hypertrophy assessment

LVH was assessed using standard transthoracic echocardiography performed according to the American Society of Echocardiography (ASE) and European Association of Cardiovascular Imaging (EACVI) guidelines [11]. Left ventricular mass (LVM) was calculated using standard M-mode/2D-derived measurements and indexed to body surface area (g/m²). LVH was defined using sex-specific guideline thresholds, as LVMI >102 g/m² in men and >88 g/m² in women, in accordance

with ASE/European Association of Cardiovascular Imaging recommendations. Sonographers were blinded to participants' HIV status and other clinical characteristic [12].

## Data collection

Sociodemographic and clinical data were collected by trained personnel using standardized instruments and entered into the Research Electronic Data Capture (REDCap) platform. Information included age, sex, marital and employment status, and medical history (HIV, hypertension, cardiovascular diseases, tuberculosis). HIV and ART details were verified through medical records.

## Statistical analysis

Data were exported from REDCap, cleaned in Microsoft Excel, and analyzed in StatCrunch. Continuous variables are presented as median (interquartile range), and categorical variables as frequency (%). Univariate associations with hypertension were assessed using chi-square or Mann–Whitney U tests, as appropriate. Variables considered for inclusion in multivariable logistic regression models were selected based on a combination of statistical significance in univariable analyses (p < 0.05) and clinical relevance informed by prior literature. Model construction was guided by epidemiological plausibility rather than automated stepwise selection procedures. The final multivariable model included age, sex, hypertension, and clinically relevant variables identified a priori, with model complexity limited to reduce overfitting. Adjusted odds ratios (AOR) with 95% confidence intervals (CI) were reported. A two-tailed p < 0.05 was considered statistically significant.

Given the limited number of LVH events in sex-stratified analyses, particularly among males, the number of covariates included in multivariable models was restricted to minimize overfitting. Events-per-variable (EPV) considerations were taken into account when constructing the final models.

## Ethical approval

Ethical approval for this study was obtained from the Mulungushi University School of Medicine and Health Sciences Research Ethics Committee (SMHS-MU REC) (Reference No. SMHS-MU3-2023-005; approval date: 9 July 2023). Institutional authorization was additionally granted by Livingstone University Teaching Hospital (LUTH) management prior to study initiation.

The approved protocol covered the entire study period (October 1, 2023, to June 1, 2024), and no extensions were required. The study was conducted at a single site (Livingstone University Teaching Hospital), and no additional site-specific approvals were necessary.

Written informed consent was obtained from all participants prior to enrollment. The study adhered to the principles of the Declaration of Helsinki (2013) and relevant national ethical guidelines. All data were anonymized prior to analysis, and no personally identifiable information was retained.

## Results

### Participants characteristics

A population (N = 333) with and without LVH was recruited. LVH was detected by echocardiography in 75 participants (22.5%). Participants with LVH were significantly older, with a median age of 59 years (IQR: 51, 65), compared to 46 years (IQR: 37, 56) in those without LVH (p < 0.0001) (Table 1). A higher proportion of females had LVH (25.9%) compared to males (14.9%; p = 0.027). Key cardiometabolic risk factors were strongly associated with LVH. Hypertensive participants had a markedly higher prevalence of LVH (45.8%) than normotensive participants (14.8%; p < 0.0001). Similarly, participants with heart failure had a very high prevalence of LVH (76.9%) compared to those without (20.0%; p < 0.0001).

**Table 1. Baseline Characteristics of the Study Population, Stratified by Left Ventricular Hypertrophy Status.**

| Variable | Median (IQR) / Frequency (%) | Left Ventricular Hypertrophy | | P value |
| --- | --- | --- | --- | --- |
| | | Yes = 75 (22.5) | No = 258 (77.5) | |
| **Age, Years** | 49 (40, 59) | 59 (51, 65) | 46 (37, 56) | **<0.0001** |
| **Sex** | | | | |
| Male | 101 (30.3) | 15 (14.9) | 86 (85.1) | **0.027** |
| Female | 232 (69.7) | 60 (25.9) | 172 (74.1) | |
| **Marital Status** | | | | |
| Married | 142 (42.8) | 37 (26.1) | 105 (73.9) | 0.192 |
| Un-married | 190 (57.2) | 38 (20.0) | 152 (80.0) | |
| **Employment Status** | | | | |
| Employed | 110 (33.3) | 23 (20.9) | 87 (79.1) | 0.577 |
| Un-employed | 220 (66.7) | 52 (23.6) | 168 (76.4) | |
| **HIV Status** | | | | |
| Positive | 244 (73.3) | 52 (21.3) | 192 (78.7) | 0.381 |
| Negative | 89 (26.7) | 23 (25.8) | 66 (74.2) | |
| **ART Regimen** | | | | |
| NNRTI+NRTI | 4 (2.6) | 0 (0) | 4 (100) | 0.084 |
| INSTI (TLD or TafED) | 149 (95.5) | 30 (20.1) | 119 (79.9) | |
| PIs | 3 (1.9) | 2 (66.7) | 1 (33.3) | |
| **Hypertension** | | | | |
| Hypertensive | 83 (24.9) | 38 (45.8) | 45 (54.2) | **<0.0001** |
| Normotensive | 250 (75.1) | 37 (14.8) | 213 (85.2) | |
| **Heart Failure** | | | | |
| Yes | 13 (4.0) | 10 (76.9) | 3 (23.1) | **<0.0001** |
| No | 315 (96.0) | 63 (20.0) | 252 (80.0) | |
| **Peripheral neuropathy presence** | | | | |
| Yes | 160 (51.3) | 44 (27.5) | 116 (72.5) | **0.019** |
| No | 152 (48.7) | 25 (16.5) | 127 (83.5) | |
| **Smoking** | | | | |
| Yes | 20 (6.5) | 6 (30.0) | 14 (70.0) | 0.427 |
| No | 288 (93.5) | 66 (22.9) | 222 (77.1) | |
| **History of TB** | | | | |
| Yes | 29 (12.9) | 11 (37.9) | 18 (62.1) | **0.012** |
| No | 196 (87.1) | 35 (17.9) | 161 (82.1) | |
| **BMI Kg/m²** | 24.5 (20.7, 28.7) | 26.2 (23.0, 30.2) | 23.8 (20.2, 28.3) | **0.0013** |
| **Waist Circumference (cm)** | 83.7 (74.0, 95.1) | 89.5 (80.0, 102.0) | 81.0 (73.0, 92.2) | **<0.0001** |
| **SSBP Delta** | 2.0 (0.0, 9.0) | 6.0 (1.0, 12.0) | 2.0 (0.0, 7.0) | **0.0002** |
| **Duration on ART (Months)** | 11 (8, 17) | 15 (10, 19) | 10 (7, 16) | **0.0455** |
| **Fasting Glucose (mmol/l)** | 4.8 (4.3, 5.4) | 5.0 (4.5, 5.6) | 4.8 (4.3, 5.4) | **0.0248** |
| **VLDL (mmol/l)** | 0.4 (0.3, 0.5) | 0.4 (0.3, 0.6) | 0.3 (0.3, 0.5) | **0.0426** |
| **Non HDL Cholesterol (mmol/l)** | 3.2 (2.5, 3.9) | 3.5 (3.0, 4.5) | 3.0 (2.5, 3.8) | **0.0012** |
| **Total Cholesterol (mmol/l)** | 4.4 (3.6, 5.1) | 4.7 (3.9, 5.7) | 4.3 (3.6, 5.0) | **0.0038** |
| **FMD test baseline arterial (mm)** | 3.7 (3.3, 4.2) | 4.1 (3.7, 4.8) | 3.6 (3.2, 4.1) | **0.0007** |
| **LDL Cholesterol (mmol/l)** | 3.0 (2.4, 3.6) | 3.3 (2.6, 4.2) | 2.9 (2.3, 3.6) | **0.0042** |
| **Cholesterol HDL Ratio** | 3.6 (3.0, 4.5) | 4.0 (3.4, 5.0) | 3.5 (2.9, 4.4) | **0.017** |
| **Framingham risk score (%)** | 2.6 (0.6, 6.3) | 4.5 (2.4, 6.8) | 1.7 (0.4, 4.9) | **0.0033** |
| **Atherosclerotic cardiovascular disease risk score (%)** | 3.6 (1.5, 7.5) | 5.5 (3.1, 9.1) | 2.9 (0.9, 6.5) | **0.0037** |

*(Continued)*

**Table 1.** (Continued)

| Variable | Median (IQR) / Frequency (%) | Left Ventricular Hypertrophy | | P value |
| --- | --- | --- | --- | --- |
| | | Yes = 75 (22.5) | No = 258 (77.5) | |
| **BNP (pg/ml)** | 3.0 (1.3, 23.7) | 4.4 (1.3, 25.6) | 1.3 (1.3, 23.7) | **0.0481** |

**Footnote**: Data are presented as Median (Interquartile Range) or Frequency (%). P values are derived from Mann-Whitney U tests for continuous variables and Chi-square (or Fisher's exact) tests for categorical variables. **Abbreviations**: IQR, Interquartile Range; ART, Antiretroviral Therapy; NNRTI, Non-Nucleoside Reverse Transcriptase Inhibitor; NRTI, Nucleoside Reverse Transcriptase Inhibitor; INSTI, Integrase Strand Transfer Inhibitor; TLD, Tenofovir/ Lamivudine/ Dolutegravir; TafED, Tenofovir alafenamide/ Emtricitabine/ Dolutegravir; PI, Protease Inhibitor; BMI, Body Mass Index; SSBP, Salt Sensitivity Blood Pressure; VLDL, Very Low-Density Lipoprotein; HDL, High-Density Lipoprotein; LDL, Low-Density Lipoprotein; FMD, Flow-Mediated Dilation; BNP, B-type Natriuretic Peptide; TB, Tuberculosis. SSBP Delta, is defined as change in systolic blood pressure during salt sensitivity testing.

Measures of adiposity, including BMI and waist circumference, were also significantly higher in the LVH group (p = 0.0013 and p < 0.0001, respectively). Clinical and biochemical markers further differentiated the groups. Participants with LVH had a longer median duration on antiretroviral therapy (ART) (15 vs. 10 months, p = 0.0455), a greater rise in systolic blood pressure (SSBP Delta: 6.0 vs. 2.0 mmHg, p = 0.0002), and higher fasting glucose levels (p = 0.0248). The LVH group exhibited a more atherogenic lipid profile, with significantly higher levels of VLDL, non-HDL cholesterol, total cholesterol, LDL cholesterol, and cholesterol-HDL ratio (all p < 0.05). Endothelial function, assessed by baseline arterial diameter in the FMD test, was also higher in the LVH group (p = 0.0007). Accordingly, calculated cardiovascular risk scores (Framingham risk score and atherosclerotic cardiovascular disease risk score) were significantly elevated in participants with LVH (p = 0.0033 and p = 0.0037, respectively). BNP levels were also higher in the LVH group (p = 0.0481). Other factors showing a significant association with LVH included a history of peripheral neuropathy (27.5% vs. 16.5%, p = 0.019) and a history of tuberculosis (37.9% vs. 17.9%, p = 0.012). No statistically significant differences were observed between the groups regarding marital status, employment status, HIV status (positive vs. negative), current ART regimen among those on treatment, or smoking status.

## Correlogram correlation matrix of clinical and metabolic parameters

The heatmap displays Pearson correlation coefficients (r) between various clinical and demographic variables studied in the cohort. LVH: Left Ventricular Hypertrophy (primary outcome); SSBPdelta: Systolic Blood Pressure Delta; BMI: Body Mass Index; WC: Waist Circumference; ART: Antiretroviral Therapy; FMD: Flow-Mediated Dilation; HDL: High-Density Lipoprotein; LDL: Low-Density Lipoprotein; VLDL: Very-Low-Density Lipoprotein; ASCVD: Atherosclerotic Cardiovascular Disease Risk; BNP: B-type Natriuretic Peptide (Fig 1). Positive correlations (r > 0) are indicated in progressively darker shades of one color (e.g., blue), while negative correlations (r < 0) are indicated in shades of another (e.g., red), with the intensity corresponding to the strength of the association. Coefficients are presented in the cells.

## Left ventricular hypertrophy vs no-left ventricular hypertrophy

Comparisons are shown between participants without LVH (No-LVH) and those with LVH across multiple domains (Fig 2). (A) Atherosclerotic cardiovascular disease (ASCVD) risk (%). (B) Body mass index (BMI, kg/m$^2$). (C) Waist circumference (cm). (D) Salt sensitivity blood pressure (SSBP, mmHg). (E) Duration on antiretroviral therapy (ART, months). (F) Fasting plasma glucose (mmol/L). (G) Total cholesterol (mmol/L). (H) Very-low-density lipoprotein (VLDL, mmol/L). (I) Non-HDL cholesterol (mmol/L). (J) Low-density lipoprotein (LDL) cholesterol (mmol/L). (K) Total cholesterol to HDL cholesterol ratio. (L) Framingham risk score (%). (M) ASCVD risk (%) using an alternative risk categorization. (N) B-type natriuretic peptide (BNP, pg/mL) (Fig 2). All measured variables demonstrated statistically significant differences between the No-LVH and LVH groups except BNP, which did not differ significantly between groups. Data are shown as individual data points with

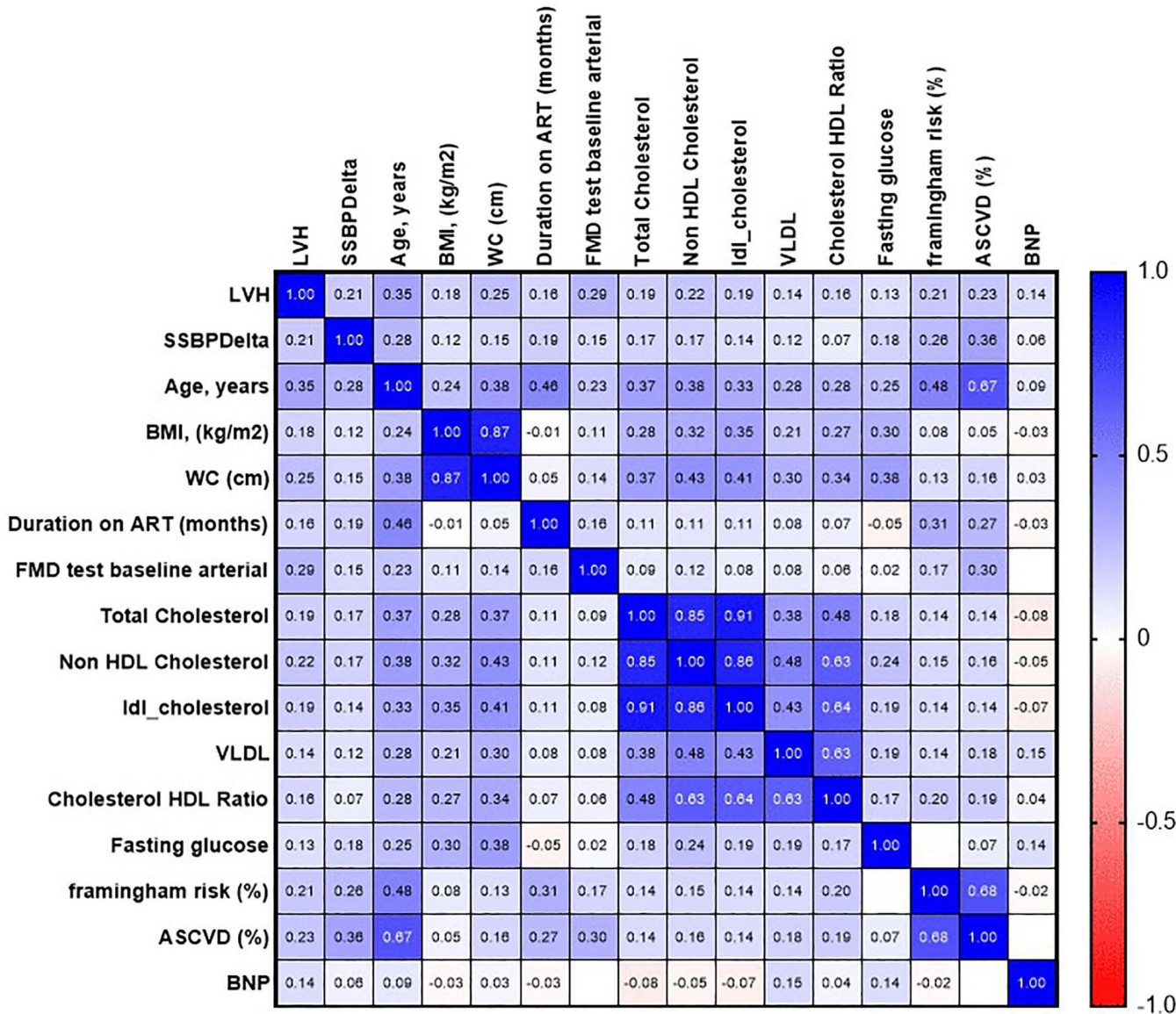

**Fig 1. Correlogram.**

bars representing central tendency and error bars indicating variability. Statistical significance is denoted as p<0.05 (*), p<0.01 (**), p<0.001 (***), and p<0.0001 (****); ns, not significant (Fig 2).

## Multivariable logistic regression for factor associated with Left Ventricular Hypertrophy

In the univariable analysis, several factors were significantly associated with LVH. Advancing age (OR: 1.07, 95% CI: 1.05-1.10; p<0.0001), female sex (OR: 2.0, 95% CI: 1.07-3.72; p=0.029), and the presence of hypertension (OR: 4.86, 95% CI: 2.78-8.47; p<0.0001) were all significant predictors (Table 2). Heart failure showed the strongest univariable association (OR: 13.3, 95% CI: 3.56-49.8; p<0.0001). Several cardiometabolic parameters, including higher BMI, waist circumference, Salt Sensitivity blood pressure change (SSBP Delta), fasting glucose, and atherogenic lipid fractions

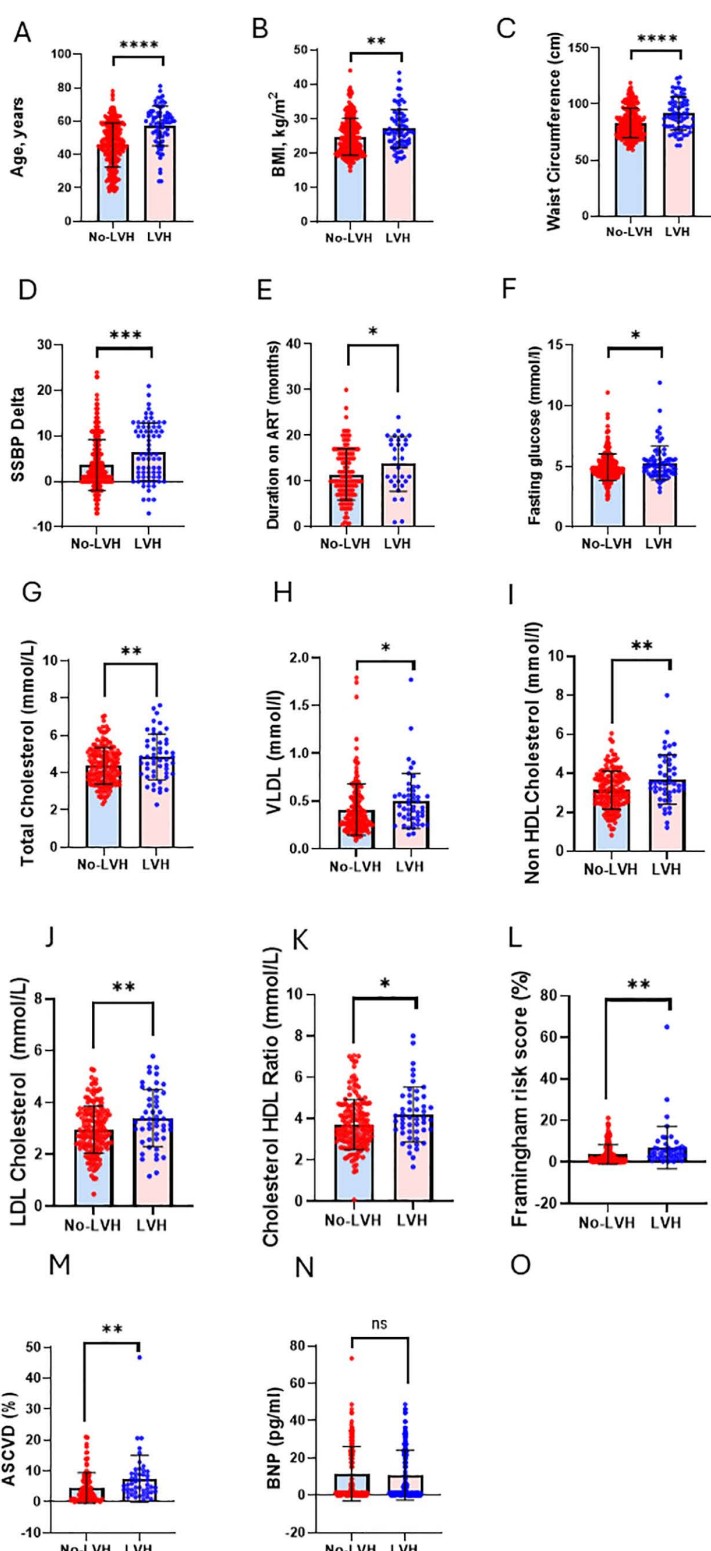

**Fig 2. Left Ventricular Hypertrophy vs No-Left Ventricular Hypertrophy.**

**Table 2. Multivariable logistic regression for factor associated with Left Ventricular Hypertrophy.**

| Variable | OR (95% CI) | P value | AOR (95% CI) | P value |
|---|---|---|---|---|
| **Age, Years** | 1.07 (1.05, 1.10) | **<0.0001** | 1.06 (1.03, 1.09) | **<0.0001** |
| **Sex** | | | | |
| Male | Ref | | Ref | |
| Female | 2.0 (1.07, 3.72) | **0.029** | 2.31 (1.13, 4.68) | **0.020** |
| **Hypertension** | | | | |
| Normotensive | Ref | | Ref | |
| Hypertensive | 4.86 (2.78, 8.47)) | **<0.0001** | 2.52 (1.36, 4.66) | **0.003** |
| **Heart Failure** | | | | |
| No | Ref | | Ref | |
| Yes | 13.3 (3.56, 49.8) | **<0.0001** | 7.50 (1.85, 30.3) | **0.005** |
| **Peripheral neuropathy presence** | | | | |
| No | Ref | | Ref | |
| Yes | 1.92 (1.10, 3.34) | **0.020** | 0.92 (0.48, 1.75) | 0.801 |
| **History of TB** | | | | |
| No | Ref | | Ref | |
| Yes | 2.81 (1.22, 6.47) | **0.015** | 1.39 (0.54, 3.55) | 0.482 |
| **BMI (Kg/m²)** | 1.07 (1.02, 1.12) | **0.002** | 1.03 (0.98, 1.09) | 0.181 |
| **Waist Circumference (cm)** | 1.04 (1.02, 1.06) | **<0.0001** | 1.02 (0.98, 1.06) | 0.279 |
| **SSBP Delta** | 1.08 (1.03, 1.13) | **<0.0001** | 1.01 (0.96, 1.07) | 0.498 |
| **Duration on ART (Months)** | 1.07 (1.00, 1.15) | **0.049** | 0.99 (0.91, 1.07) | 0.836 |
| **Fasting Glucose (mmol/l)** | 1.26 (1.02, 1.55) | **0.029** | 0.98 (0.78, 1.23) | 0.880 |
| **VLDL (mmol/l** | 2.88 (0.99, 8.33) | 0.050 | 1.45 (0.42, 5.04) | 0.552 |
| **Non HDL Cholesterol (mmol/l)** | 1.62 (1.19, 2.21) | **0.002** | 1.23 (0.85, 1.76) | 0.258 |
| **Total Cholesterol (mmol/l)** | 1.53 (1.14, 2.06) | **0.005** | 1.13 (0.80, 1.59) | 0.482 |
| **FMD test baseline arterial (mm)** | 2.16 (1.29, 3.61) | **0.003** | 1.70 (0.99, 2.91) | 0.052 |
| **LDL Cholesterol (mmol/l)** | 1.60 (1.15, 2.24) | **0.005** | 1.19 (0.81, 1.76) | 0.359 |
| **Cholesterol HDL Ratio** | 1.34 (1.05, 1.73) | **0.019** | 1.19 (0.89, 1.59) | 0.234 |
| **Framingham risk (%)** | 1.07 (1.01, 1.14) | **0.013** | 0.99 (0.94, 1.05) | 0.860 |
| **ASCVD (%)** | 1.08 (1.02, 1.15) | **0.009** | 0.97 (0.89, 1.05) | 0.489 |
| **BNP (pg/ml)** | 1.00 (0.99, 1.00) | 0.748 | 1.00 (0.99, 1.00) | 0.568 |

(non-HDL cholesterol, total cholesterol, LDL cholesterol, and cholesterol-HDL ratio), were also significant predictors. Notably, a history of peripheral neuropathy (OR: 1.92, p = 0.020) and tuberculosis (OR: 2.81, p = 0.015), as well as a longer duration on ART (OR: 1.07, p = 0.049), were associated with LVH in univariable models. Increased baseline arterial diameter (FMD test) and higher Framingham and ASCVD risk scores were also significant.

In the multivariable model, which adjusted for other covariates using age, bmi and hypertension, only four factors remained independently associated with LVH: age (AOR: 1.06, 95% CI: 1.03-1.09; p < 0.0001), female sex (AOR: 2.31, 95% CI: 1.13-4.68; p = 0.020), hypertension (AOR: 2.52, 95% CI: 1.36-4.66; p = 0.003), and heart failure (AOR: 7.50, 95% CI: 1.85-30.3; p = 0.005) (Table 2). The associations observed in the univariable analysis for other variables, including peripheral neuropathy, history of TB, adiposity measures, lipid profiles, and cardiovascular risk scores, were attenuated and no longer statistically significant in the adjusted model. The association with baseline arterial diameter approached, but did not reach, statistical significance (AOR: 1.70, 95% CI: 0.99-2.91; p = 0.052).

## Forest plot of factors independently associated with left ventricular hypertrophy

This forest plot (Fig 3) displays the results of the multivariable logistic regression analysis identifying factors independently associated with LVH. The model was adjusted for age, body mass index, and hypertension. Solid circles represent the AOR point estimate for each factor, with the size of the marker corresponding to the precision of the estimate. Horizontal lines depict the 95% CIs. The vertical dashed line indicates the line of null effect (AOR = 1). Associations are considered statistically significant (p < 0.05) when the 95% CI does not cross this line. Four factors were significant independent predictors of LVH: advancing age, female sex, hypertension, and a history of heart failure. Baseline arterial diameter (from FMD testing) showed a strong positive association that approached, but did not reach, conventional statistical significance (AOR: 1.70, 95% CI: 0.99-2.91; p = 0.052).

## Baseline characteristics and associations with left ventricular hypertrophy, stratified by sex

Among males, significant differences between those with and without LVH were observed for age (median 59 vs. 48 years, p = 0.002), hypertension (34.6% vs. 8.0%, p = 0.0008), heart failure (80.0% vs. 11.5%, p < 0.0001), a history of TB (36.4% vs. 10.6%, p = 0.024), the Salt sensitivity blood pressure change (SSBP Delta) (median 6 vs. 2 mmHg, p = 0.003), duration on ART (median 18.5 vs. 12 months, p = 0.011), and both the Framingham (median 7.5% vs. 4.6%, p = 0.041) and ASCVD risk scores (median 7.9% vs. 4.9%, p = 0.043) (Table 3). Notably, BMI, lipid profiles, and baseline arterial diameter were not significantly different between male groups.

Among females, significant differences between those with and without LVH were observed for age (median 60 vs. 46 years, p < 0.0001), hypertension (50.9% vs. 17.7%, p < 0.0001), heart failure (75.0% vs. 23.7%, p = 0.001), peripheral neuropathy (31.3% vs. 18.7%, p = 0.032), BMI (median 26.3 vs. 25.1 kg/m², p = 0.021), waist circumference (median 92 vs. 83 cm, p = 0.0001), SSBP Delta (median 6 vs. 0 mmHg, p = 0.007), and multiple lipid parameters (VLDL, non-HDL cholesterol, total cholesterol, LDL cholesterol, cholesterol-HDL ratio, all p < 0.05) (Table 3). Furthermore, baseline arterial

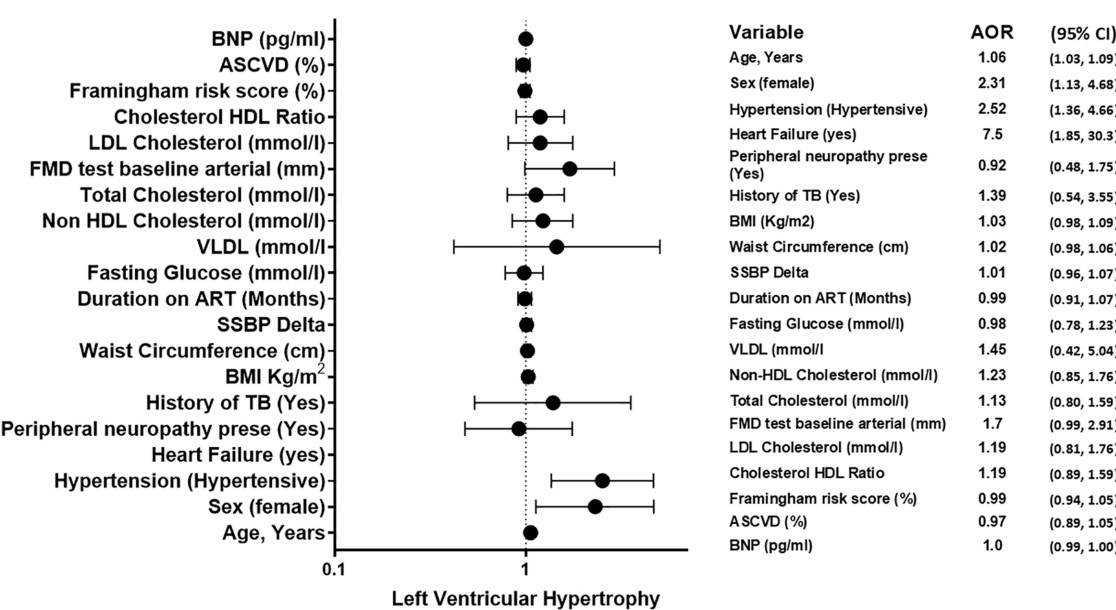

| Variable | AOR | (95% CI) |
|---|---|---|
| Age, Years | 1.06 | (1.03, 1.09) |
| Sex (female) | 2.31 | (1.13, 4.68) |
| Hypertension (Hypertensive) | 2.52 | (1.36, 4.66) |
| Heart Failure (yes) | 7.5 | (1.85, 30.3) |
| Peripheral neuropathy prese (Yes) | 0.92 | (0.48, 1.75) |
| History of TB (Yes) | 1.39 | (0.54, 3.55) |
| BMI (Kg/m2) | 1.03 | (0.98, 1.09) |
| Waist Circumference (cm) | 1.02 | (0.98, 1.06) |
| SSBP Delta | 1.01 | (0.96, 1.07) |
| Duration on ART (Months) | 0.99 | (0.91, 1.07) |
| Fasting Glucose (mmol/l) | 0.98 | (0.78, 1.23) |
| VLDL (mmol/l | 1.45 | (0.42, 5.04) |
| Non-HDL Cholesterol (mmol/l) | 1.23 | (0.85, 1.76) |
| Total Cholesterol (mmol/l) | 1.13 | (0.80, 1.59) |
| FMD test baseline arterial (mm) | 1.7 | (0.99, 2.91) |
| LDL Cholesterol (mmol/l) | 1.19 | (0.81, 1.76) |
| Cholesterol HDL Ratio | 1.19 | (0.89, 1.59) |
| Framingham risk score (%) | 0.99 | (0.94, 1.05) |
| ASCVD (%) | 0.97 | (0.89, 1.05) |
| BNP (pg/ml) | 1.0 | (0.99, 1.00) |

**Fig 3. Forest Plot for factor associated with Left Ventricular Hypertrophy.**

**Table 3.** Baseline Characteristics and Associations with Left Ventricular Hypertrophy, Stratified by Sex.

| Variable | Male | | | | Female | | | |
|---|---|---|---|---|---|---|---|---|
| | Median (IQR) / Frequency (%) | Left Ventricular Hypertrophy | | P value | Median (IQR) / Frequency (%) | Left Ventricular Hypertrophy | | P value |
| | | Yes = 15 (14.9) | No = 86 (85.1) | | | Yes = 60 (25.9) | No = 172 (74.1) | |
| **Age, Years** | 50 (42, 59) | 59 (52, 63) | 48 (41, 56) | **0.0022** | 48 (40, 59) | 60 (50, 65) | 46 (36, 55) | **<0.0001** |
| **Marital Status** | | | | | | | | |
| *Married* | 71 (70.3) | 13 (18.3) | 58 (81.7) | 0.1354 | 71 (30.7) | 24 (33.8) | 47 (66.2) | 0.071 |
| *Un-married* | 30 (29.7) | 2 (6.7) | 28 (93.3) | | 160 (69.3) | 36 (22.5) | 124 (77.5) | |
| **Employment Status** | | | | | | | | |
| *Employed* | 48 (47.5) | 8 (16.7) | 40 (83.3) | 0.6295 | 62 (27.1) | 15 (24.2) | 47 (75.8) | 0.674 |
| *Un-employed* | 53 (52.5) | 7 (13.2) | 46 (86.8) | | 167 (72.9) | 45 (27.0) | 122 (73.0) | |
| **HIV Status** | | | | | | | | |
| *Positive* | 65 (64.4) | 8 (12.3) | 57 (87.7) | 0.339 | 179 (77.2) | 44 (24.6) | 135 (75.4) | 0.413 |
| *Negative* | 36 (35.6) | 7 (19.4) | 29 (80.6) | | 53 (22.8) | 16 (30.2) | 37 (69.8) | |
| **Hypertension** | | | | | | | | |
| *Hypertensive* | 26 (25.7) | 9 (34.6) | 17 (65.4) | **0.0008** | 57 (24.6) | 29 (50.9) | 28 (49.1) | **<0.0001** |
| *Normotensive* | 75 (74.3) | 6 (8.0) | 69 (92.0) | | 175 (75.4) | 31 (17.7) | 144 (82.3) | |
| **Heart Failure** | | | | | | | | |
| *Yes* | 5 (5.0) | 4 (80.0) | 1 (20.0) | **<0.0001** | 8 (3.5) | 6 (75.0) | 2 (25.0) | **0.001** |
| *No* | 96 (95.0) | 11 (11.5) | 85 (88.5) | | 219 (96.5) | 52 (23.7) | 167 (76.3) | |
| **Peripheral neuropathy presence** | | | | | | | | |
| *Yes* | 48 (51.6) | 9 (18.8) | 39 (81.2) | 0.3085 | 112 (51.1) | 35 (31.3) | 77 (68.8) | **0.032** |
| *No* | 45 (48.4) | 5 (11.1) | 40 (88.9) | | 107 (48.9) | 20 (18.7) | 87 (81.3) | |
| **Smoking** | | | | | | | | |
| *Yes* | 10 (10.6) | 1 (10.0) | 9 (90.0) | 0.591 | 10 (4.7) | 5 (50.0) | 5 (50.0) | 0.135 |
| *No* | 84 (89.4) | 14 (16.7) | 70 (83.3) | | 204 (95.3) | 52 (25.5) | 152 (74.5) | |
| **History of TB** | | | | | | | | |
| *Yes* | 11 (14.3) | 4 (36.4) | 7 (63.6) | **0.0237** | 18 (12.2) | 7 (38.9) | 11 (61.1) | 0.104 |
| *No* | 66 (85.7) | 7 (10.6) | 59 (89.4) | | 130 (87.8) | 28 (21.5) | 102 (78.5) | |
| **BMI Kg/m²** | 21.9 (19.8, 26.3) | 23.0 (20.4, 27.6) | 21.5 (19.8, 26.3) | 0.3932 | 25.9 (21.8, 29.9) | 26.3 (23.6, 30.8) | 25.1 (21.2, 29.4) | **0.021** |
| **Waist Circumference (cm)** | 78 (72.2, 89.0) | 79.5 (74, 100) | 77.5 (72, 88) | 0.4267 | 87 (75.0, 97.0) | 92 (81.1, 104.1) | 83 (73.3, 94.0) | **0.0001** |
| **SSBP Delta** | | 6 (1, 14) | 2 (0, 6) | **0.0032** | 2 (0, 9) | 6 (1, 12) | 0 (0, 7) | **0.0066** |
| **Duration on ART (Months)** | 14 (7, 17) | 18.5 (17, 20) | 12 (7, 17) | **0.0112** | 10 (8, 16) | 11 (9, 18) | 10 (8, 15) | 0.2868 |
| **Fasting Glucose (mmol/l)** | 4.8 (4.2, 5.3) | 5.0 (4.6, 5.4) | 4.7 (4.2, 5.3) | 0.0739 | 4.8 (4.4, 5.5) | 5.0 (4.4, 5.6) | 4.8 (4.4, 5.4) | 0.1404 |
| **VLDL (mmol/l)** | 0.3 (0.3, 0.5) | 0.4 (0.3, 0.6) | 0.3 (0.3, 0.6) | 0.7654 | 0.4 (0.3, 0.6) | 0.5 (0.3, 0.6) | 0.3 (0.2, 0.5) | **0.0193** |
| **Non HDL Cholesterol (mmol/l)** | 2.8 (2.3, 3.7) | 3.0 (2.2, 3.3) | 2.8 (2.4, 3.7) | 0.8109 | 3.4 (2.7, 4.0) | 3.6 (3.2, 4.5) | 3.1 (2.5, 3.7) | **0.0011** |
| **Total Cholesterol (mmol/l)** | 4.1 (3.5, 4.9) | 4.1 (3.1, 5.3) | 4.1 (3.5, 4.9) | 0.767 | 4.5 (3.8, 5.3) | 4.8 (4.2, 5.8) | 4.5 (3.6, 5.1) | **0.01** |
| **FMD test baseline arterial (mm)** | 3.8 (3.4, 4.4) | 4.2 (3.8, 5.1) | 3.6 (3.4, 4.3) | 0.0937 | 3.7 (4.1) | 3.9 (3.7, 4.5) | 3.5 (3.1, 4.0) | **0.002** |
| **LDL Cholesterol (mmol/l)** | 2.7 (2.2, 3.4) | 2.8 (2.0, 3.2) | 2.7 (2.2, 3.5) | 0.967 | 3.2 (2.6, 4.0) | 3.4 (3.0, 4.5) | 3.1 (2.5, 3.7) | **0.0102** |

*(Continued)*

**Table 3.** (Continued)

| Variable | Male | | | | Female | | | |
|---|---|---|---|---|---|---|---|---|
| | Median (IQR) / Frequency (%) | Left Ventricular Hypertrophy | | P value | Median (IQR) / Frequency (%) | Left Ventricular Hypertrophy | | P value |
| | | Yes = 15 (14.9) | No = 86 (85.1) | | | Yes = 60 (25.9) | No = 172 (74.1) | |
| Cholesterol HDL Ratio | 3.4 (2.9, 4.2) | 3.4 (2.8, 3.9) | 3.4 (2.9, 4.4) | 0.3657 | 3.8 (3.0, 4.6) | 4.2 (3.6, 5.1) | 3.5 (2.9, 4.4) | **0.0022** |
| Framingham risk (%) | 4.9 (2.7, 9.5) | 7.5 (4.2, 12.2) | 4.6 (2.1, 9.3) | **0.0411** | 1.4 (0.3, 3.7) | 3.4 (1.6, 6.5) | 0.7 (0.1, 2.3) | **0.0036** |
| ASCVD (%) | 5.8 (2.7, 10.1) | 7.9 (4.6, 12.8) | 4.9 (2.5, 9.0) | **0.0428** | 3 (0.9, 6.1) | 5.2 (3.1, 8.9) | 1.9 (0.7, 4.1) | **0.0006** |
| BNP (pg/ml) | 1.3 (1.3, 21.7) | 2.2 (0.8, 5) | 1.3 (1.3, 21.7) | 0.2881 | 3.0 (1.3, 25.6) | 5.6 (1.3, 27.5) | 1.3 (1.3, 23.7) | 0.0723 |

**Abbreviations:** LVH; Left ventricular hypertrophy, IQR; Interquartile range, ART; Antiretroviral therapy, HIV; Human immunodeficiency virus, TB; Tuberculosis, BMI; Body mass index, kg/m²; Kilograms per square metre, cm; Centimetres, SBP; Systolic blood pressure, SSBP Delta; Change (difference) in systolic blood pressure, mmHg;Millimetres of mercury, mmol/L; Millimoles per litre, VLDL; Very low-density lipoprotein cholesterol, LDL;Low-density lipoprotein cholesterol, HDL; High-density lipoprotein cholesterol, Non-HDL cholesterol; Total cholesterol minus HDL cholesterol, FMD; Flow-mediated dilation, ASCVD; Atherosclerotic cardiovascular disease, BNP; B-type natriuretic peptide, pg/mL; Picograms per millilitre, Ref.; Reference category, p value; Probability value.

diameter (3.9 vs. 3.5 mm, p = 0.002) and cardiovascular risk scores (Framingham: 3.4% vs. 0.7%, p = 0.004; ASCVD: 5.2% vs. 1.9%, p = 0.001) were significantly higher in females with LVH.

## Sex-stratified multivariable logistic regression for factors associated with left ventricular hypertrophy

Table 4 presents the results of sex-stratified multivariable logistic regression analyses for factors independently associated with LVH. Variables that were statistically significant (p < 0.05) at univariable analysis were included in multivariable analysis. The adjusted models revealed distinct predictors for males and females.

In males, three factors were independently associated with LVH after adjustment: increasing age (AOR: 1.06 per year, 95% CI: 1.00-1.13; p = 0.030), hypertension, and heart failure. The odds of LVH were significantly higher in males with hypertension (AOR: 3.70, 95% CI: 1.03-13.2; p = 0.044) compared to normotensives (Table 4). The association with heart failure was particularly strong (AOR: 18.3, 95% CI: 1.60-209.0; p = 0.019). History of TB, adiposity measures (BMI), hemodynamic factors (SSBP Delta), ART duration, and cardiovascular risk scores were not independent predictors in the final male model.

In females, a broader set of factors remained independently significant. Increasing age was a strong predictor (AOR: 1.06 per year, 95% CI: 1.03-1.09; p < 0.0001), as was hypertension (AOR: 2.50, 95% CI: 1.21-5.18; p = 0.013) (Table 4). Greater waist circumference (AOR: 1.05 per cm, 95% CI: 1.00-1.10; p = 0.047) and a higher cholesterol-HDL ratio (AOR: 1.53, 95% CI: 1.02-2.30; p = 0.036) were also independent metabolic correlates of LVH. The association with heart failure showed a strong trend but did not reach statistical significance (AOR: 4.92, 95% CI: 0.87-27.6; p = 0.070). Peripheral neuropathy, other lipid fractions, baseline arterial diameter, and cardiovascular risk scores were not independently associated in the final female model.

## Forest plot for factor associated with left ventricular hypertrophy in females

Results of the sex-stratified multivariable analysis for females (Fig 4). The model identifies independent predictors of LVH, adjusted for other covariates including age, BMI, and hypertension. Solid circles represent the AOR point estimate, with horizontal lines showing the 95% CI. The vertical dashed line indicates the line of null effect (AOR = 1). In females, a broader set of factors were significant independent predictors: advancing age, hypertension, greater waist circumference, and a higher cholesterol-HDL ratio (Fig 4). The association with heart failure showed a strong, non-significant trend.

**Table 4. Multivariable sex differences.**

| Variables | Males | | Female | |
|---|---|---|---|---|
| | AOR (95% CI) | P value | AOR (95% CI) | P value |
| **Age, Years** | 1.06 (1.00, 1.13) | **0.030** | 1.06 (1.03, 1.09) | **<0.0001** |
| **Hypertension** | | | | |
| *Normotensive* | Ref | | | |
| *Hypertensive* | 3.70 (1.03, 13.2) | **0.044** | 2.50 (1.21, 5.18) | **0.013** |
| **Heart Failure** | | | | |
| *No* | Ref | | | |
| *Yes* | 18.3 (1.60, 209.0) | **0.019** | 4.92 (0.87, 27.6) | 0.070 |
| **Peripheral neuropathy presence** | | | | |
| *No* | Ref | | Ref | |
| *Yes* | – | | 0.78 (0.36, 1.67) | 0.537 |
| **History of TB** | | | | |
| *No* | Ref | | Ref | |
| *Yes* | 3.23 (0.63, 16.5) | 0.157 | – | – |
| **BMI (Kg/m²)** | 0.99 (0.86, 1.13) | 0.885 | 1.02 (0.96, 1.08) | 0.472 |
| **Waist Circumference (cm)** | – | – | 1.05 (1.00, 1.10) | **0.047** |
| **SSBP Delta** | 1.07 (0.96, 1.19) | 0.206 | 0.99 (0.93,1.06) | 0.910 |
| **Duration on ART (Months)** | 1.43 (0.91, 2.24) | 0.119 | – | – |
| **VLDL (mmol/l)** | – | – | 2.28 (0.42, 12.2) | 0.335 |
| **Non HDL Cholesterol (mmol/l)** | – | – | 1.29 (0.82, 2.04) | 0.256 |
| **Total Cholesterol (mmol/l)** | – | – | 1.06 (0.69, 1.64) | 0.770 |
| **FMD test baseline arterial (mm)** | – | – | 1.61 (0.79, 3.30) | 0.186 |
| **LDL Cholesterol (mmol/l)** | – | – | 1.20 (0.76, 1.90) | 0.411 |
| **Cholesterol HDL Ratio** | – | – | 1.53 (1.02, 2.30) | **0.036** |
| **Framingham risk score (%)** | 1.03 (0.89, 1.20) | 0.624 | 1.02 (0.94, 1.11) | 0.547 |
| **Atherosclerotic cardiovascular disease risk score (%)** | 1.05 (0.94, 1.17) | 0.357 | 0.97 (0.82, 1.14) | 0.727 |

AOR = Adjusted Odds Ratio; CI = Confidence Interval; Ref = Reference category. Each stratified model was adjusted for all variables listed in the respective sex column. A dash (-) indicates the variable was not retained in the final stepwise model (p > 0.05).

### Forest plot of factors independently associated with left ventricular hypertrophy in males

Results of the sex-stratified multivariable analysis for males (Fig 5). The model identifies independent predictors of LVH, adjusted for other covariates including age, BMI, and hypertension. Solid circles represent the AOR point estimate, with horizontal lines showing the 95% CI. The vertical dashed line indicates the line of null effect (AOR = 1). In males, three factors were significant independent predictors: advancing age, hypertension, and a history of heart failure, which showed a particularly strong association (Fig 5).

## Discussion

We observed a high burden of LVH (22.5%) among adults attending a medical clinic. In the overall cohort, increasing age, female sex, hypertension, and heart failure were independently associated with the presence of LVH. Sex-stratified analyses demonstrated distinct patterns of association (Fig 6). Among males, LVH was primarily associated with advancing age, hypertension, and heart failure. In contrast, in females, LVH was associated not only with age and hypertension but also with markers of metabolic risk, including increased waist circumference and an elevated cholesterol-to–HDL ratio

PLOS Global Public Health

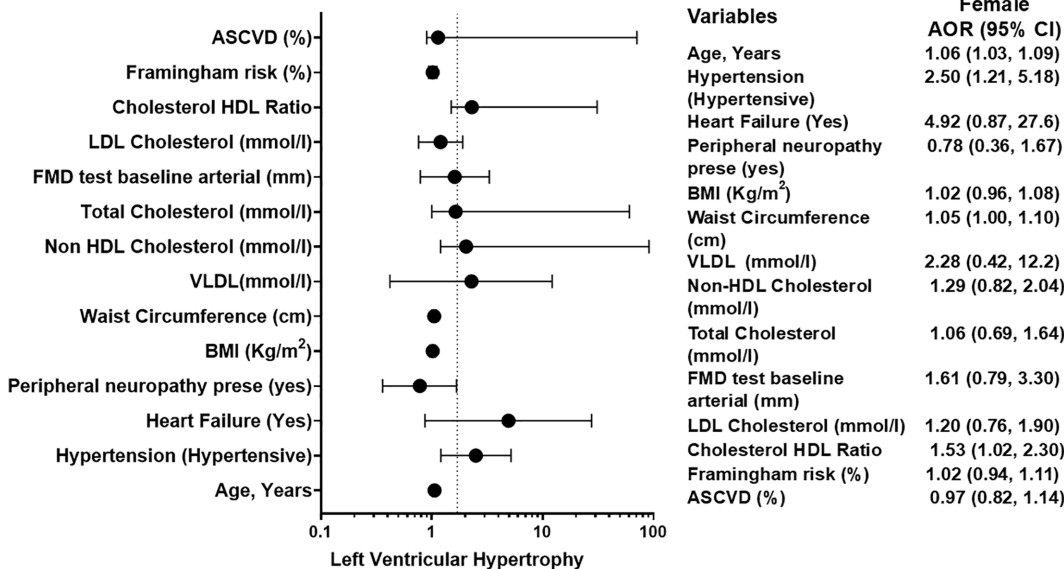

**Fig 4. Forest Plot for factor associated with Left Ventricular Hypertrophy in Females.**

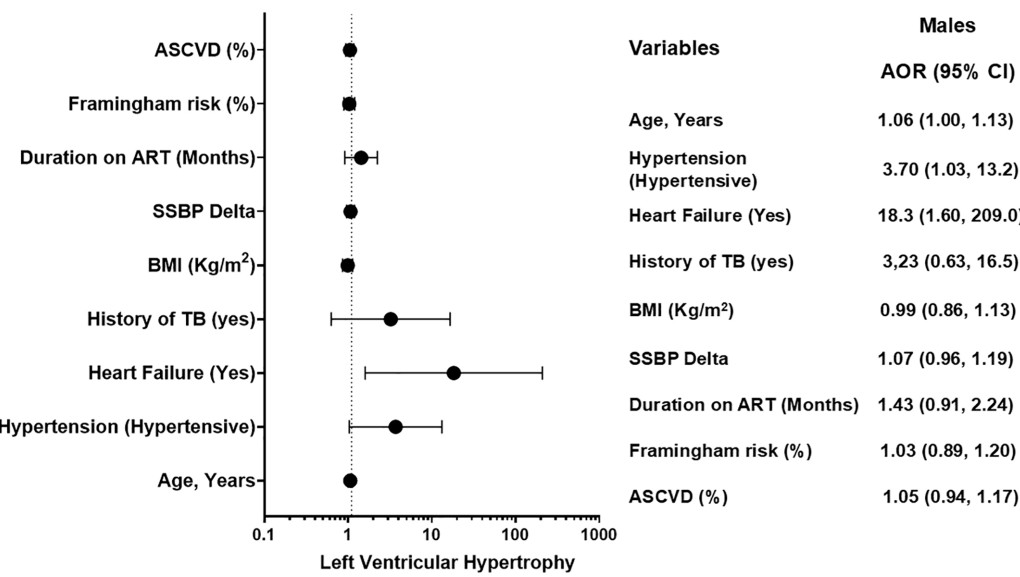

**Fig 5. Forest Plot for factor associated with Left Ventricular Hypertrophy in Males.**

(Fig 6). Collectively, these findings indicate that LVH in PLWH is associated with both conventional hemodynamic factors and sex-specific metabolic profiles.

Fig 6 summarizes the key sex-specific patterns of left ventricular hypertrophy (LVH) observed in this study. Overall LVH prevalence was 22.5%, with higher prevalence in females (25.9%) than males (14.9%). LVH in males was associated with hemodynamic factors (hypertension, heart failure), whereas in females both hemodynamic and metabolic factors

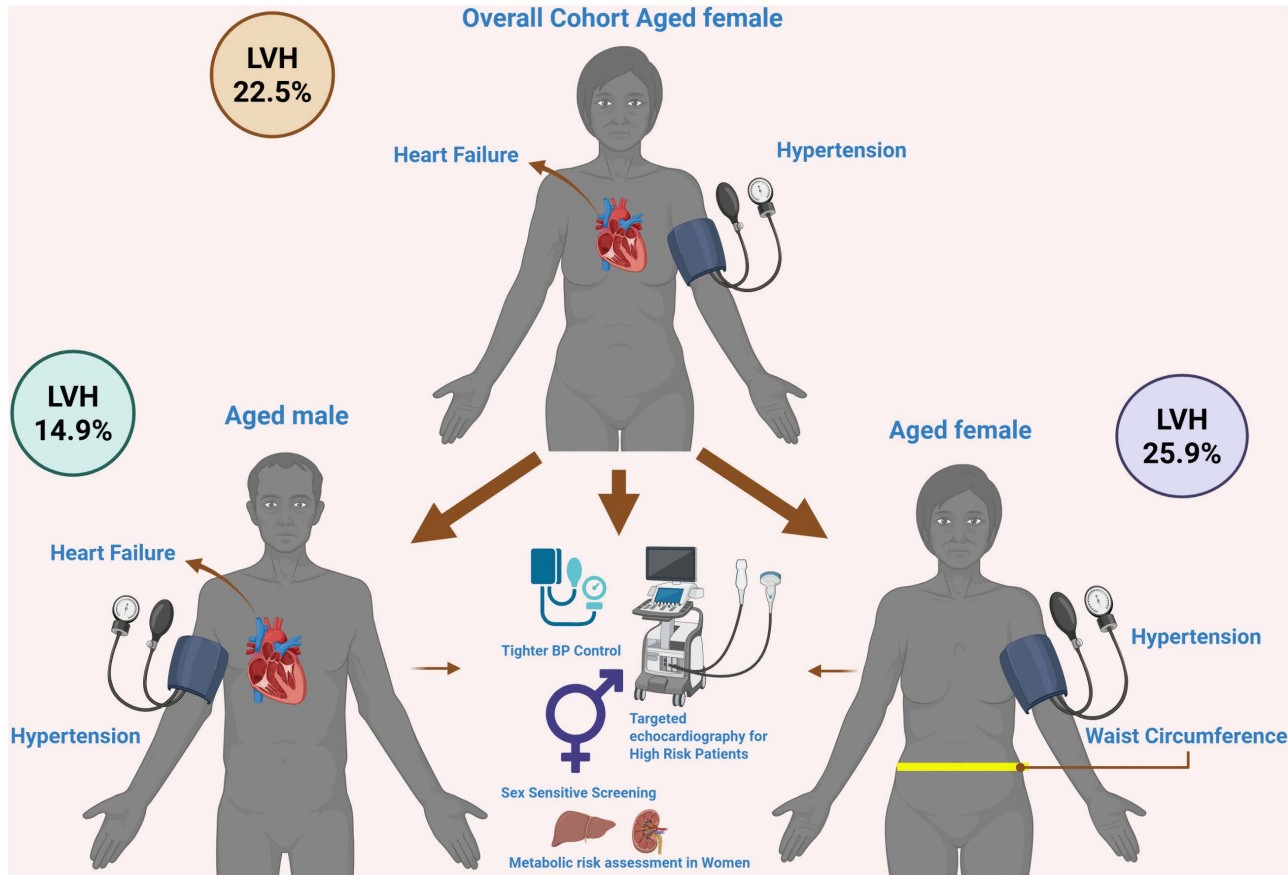

**Fig 6. Sex-specific patterns and clinical implications of left ventricular hypertrophy in a clinic-attending cohort.**

(including waist circumference) were implicated. The figure highlights implications for sex-sensitive screening, blood pressure control, and targeted echocardiography (Fig 6).

The prevalence of LVH observed in our cohort is broadly consistent with findings from studies conducted in comparable settings within sub-Saharan Africa ranging from 9 – 30% [3,13]. Echocardiographic screening studies among adults living with HIV in southern Africa have consistently demonstrated a substantial burden of LVH, which is strongly associated with elevated blood pressure and increased body mass index, highlighting the contribution of chronic hemodynamic load and metabolic stress to adverse cardiac remodeling in this population. Systematic echocardiographic assessment in Malawi identifies LVH as the most common structural abnormality, even in the absence of overt heart failure or major valve disease, underscoring the significance of subclinical myocardial remodeling in settings with high hypertension prevalence [14]. Other African cohorts report similar patterns of structural heart disease with varying prevalence, likely influenced by differences in population characteristics, ART coverage, and cardiovascular risk factor burden [15,16].

Our study revealed a significantly higher prevalence of LVH in females (25.9%) compared to males (14.9%), a finding that persisted as an independent association in the overall multivariable model. This aligns with emerging evidence suggesting that women, particularly those living with HIV, may be more susceptible to certain patterns of adverse cardiac remodeling [3,17]. The sex-stratified findings suggest potential differences in the relative contribution of risk factors, with hemodynamic factors appearing more prominent in males and metabolic factors in females. However, these observations should be interpreted with caution given the cross-sectional design and limited number of events in stratified analyses,

particularly among males. As such, these findings should be considered exploratory and hypothesis-generating rather than definitive. While pressure overload remains a universal driver, the female heart in this context may be additionally vulnerable to metabolic and possibly inflammatory insults, which could be exacerbated by sex-specific differences in body fat distribution, adipokine profiles, or immune activation [17–19]. The stronger metabolic signature in females could partly explain their higher LVH burden, indicating that cardiovascular risk assessment in women requires integrated evaluation beyond blood pressure alone.

The higher prevalence of LVH in women in our cohort deserves additional consideration. Beyond the independent association of female sex in the overall model, our sex-stratified analysis suggests that women may experience a broader cardiometabolic pathway to LV remodeling, characterized not only by hypertension but also by central adiposity and an adverse cholesterol-HDL profile. This interpretation is consistent with broader cardiovascular literature showing that women differ from men in left ventricular geometry, remodeling patterns, and susceptibility to hypertensive target-organ damage [19]. Prior work has shown that sex modifies the association between hemodynamic load and LV remodeling, and that women may demonstrate greater concentric remodeling or hypertrophic responses under similar pressure conditions [20,21]. In addition, central obesity has been linked to increased LV wall thickness and adverse myocardial remodeling, which is relevant because waist circumference, rather than BMI alone, remained independently associated with LVH in our female subgroup. In HIV populations, sex-related differences in cardiac stress and end-organ injury have also been reported, supporting the biological plausibility of a sex-specific remodeling phenotype [22,23].

The association between heart failure and LVH, particularly prominent in males in our cohort, reflects both the role of long-standing pressure overload and the potential for subclinical cardiac dysfunction to progress to symptomatic disease. In the general population, LVH is a well-established risk factor for heart failure with preserved ejection fraction, arrhythmias, and ischemic events; this relationship appears to extend to PLWH [24,25]. In addition to hypertension and age, metabolic abnormalities such as dyslipidemia and central adiposity are implicated in LVH among women in our cohort. This corroborates findings from clinical and epidemiologic studies that metabolic syndrome features are increasingly common in PLWH and are associated with adverse cardiac outcomes [26]. Metabolic derangements may promote myocardial fibrosis and interstitial remodeling via chronic inflammation, insulin resistance, and adipose-derived cytokine activity, processes that have been implicated in LV structural changes even in virally suppressed individuals [27,28].

Hypertension was one of the strongest and most consistent factor associated with of LVH in our study for both males and females. This aligns with broader evidence showing that hypertension is common among PLWH in sub-Saharan Africa and is a major driver of target organ damage such as LVH and eventual heart failure [29,1]. Hypertension prevalence estimates among African cohorts of PLWH vary, but numerous studies highlight the coexistence of hypertension and increased left ventricular mass in this population, suggesting that rigorous blood pressure screening and control should be prioritized within HIV care [30–32].

Interpretation of the sex-stratified findings, particularly among males, should consider the limited number of LVH cases in this subgroup. The small sample size resulted in reduced statistical power and wide confidence intervals, especially for heart failure, suggesting potential estimate instability. In contrast, the larger number of LVH cases among females allowed for more precise estimation and revealed a broader metabolic risk profile, including central adiposity and dyslipidaemia. Despite these limitations, the observed sex-specific patterns are biologically plausible and consistent with existing literature, supporting the hypothesis that LVH pathophysiology may differ between men and women living with HIV. Future studies with larger sex-balanced cohorts and longitudinal follow-up are required to confirm these findings and clarify causal pathways.

These findings have direct clinical implications. First, they support routine cardiovascular risk assessment in outpatient and HIV care settings that goes beyond blood pressure measurement alone. In women, simple measures such as waist circumference and lipid-related indices may help identify individuals at increased risk of subclinical cardiac remodeling even before overt cardiovascular disease becomes apparent. Second, the data support sex-sensitive prevention

strategies: for men, aggressive control of hemodynamic load and early recognition of heart failure may be especially important, whereas for women, integrated management of hypertension and metabolic risk may be required. Third, where echocardiography is not universally available, a risk-based approach using age, hypertension status, central adiposity, and clinical history may help prioritize patients for cardiac imaging and follow-up.

### Strengths and limitations

Our study possesses notable methodological strengths. We employed a comprehensive cardiovascular assessment protocol, including transthoracic echocardiography conducted by trained sonographers blinded to HIV status, adhering to established international guidelines. This rigorous approach enhances the validity of our LVH diagnosis. The cohort design, which included both PLWH and PWTH from the same clinical setting, enabled a direct comparative analysis of determinants within a population facing a high dual burden of infectious and non-communicable diseases. A principal strength of this work is the explicit sex-stratified analysis, which moves beyond reporting sex as a mere covariate to elucidate distinct risk profiles, addressing a critical gap in the literature concerning cardiovascular morbidity in sub-Saharan Africa.

Several limitations warrant consideration. First, the cross-sectional design of this study limits the ability to establish temporal relationships or infer causality between the identified factors and LVH. The observed associations should therefore be interpreted as correlational rather than causal. It is not possible to determine whether the identified risk factors preceded the development of LVH or arose as a consequence of underlying cardiovascular changes. Longitudinal studies are required to clarify temporal sequencing and causal pathways. Second, the use of a purposive, non-probability sampling method at a single tertiary hospital may affect the external validity of our prevalence estimates and risk associations, as participants may not be fully representative of all clinic attendees or community-dwelling adults in Zambia. Third, the sample size for sex-stratified analyses was limited, particularly among males, where only 15 LVH events were observed. This resulted in a low events-per-variable (EPV) ratio in the multivariable models, increasing the risk of model overfitting and instability. This is reflected in the wide confidence intervals observed for some estimates, particularly for heart failure in males. Although we limited the number of covariates included in the final models to mitigate this issue, the findings from stratified analyses should be interpreted with caution. Larger studies with adequate event numbers are needed to confirm these associations. Fourth, although we adjusted for several key demographic and cardiometabolic factors, the possibility of residual confounding remains. Important variables that may influence LVH, including duration and treatment of hypertension, renal function, and detailed HIV-related factors, were not fully captured or included in the final models. While some HIV-related variables such as ART regimen and duration on ART were assessed, viral load was uniformly suppressed by study design, and these factors were not independently associated with LVH in adjusted analyses. Additionally, data on antihypertensive medication use and duration of hypertension were not systematically available. These unmeasured or incompletely measured factors may have influenced the observed associations.

Finally, prior tuberculosis relied on medical records and self-report, which are subject to potential misclassification. Furthermore, the use of sex-specific echocardiographic criteria for LVH ($102\,g/m^2$ in men, $88\,g/m^2$ in women) may have contributed to the observed higher prevalence in women. While these criteria are widely accepted and account for physiological differences in body size, they could partly explain the stronger association between female sex and LVH in our cohort. Future studies using alternative indexing methods (e.g., LV mass) may help clarify the extent of true biological vs. diagnostic differences.

### Conclusion

Left ventricular hypertrophy was common in this cohort and was primarily associated with traditional cardiometabolic risk factors, particularly hypertension and adiposity. Sex-specific patterns were observed but should be interpreted cautiously given the cross-sectional design and limited subgroup events. No independent association with HIV status was identified

in this treated population. These findings underscore the importance of cardiometabolic risk management, while longitudinal studies are needed to confirm these observations.

## Supporting information

**S1 File. Strobe checklist.**
(DOCX)

## Author contributions

**Conceptualization:** Sydney Mulamfu, David Chisompola.

**Data curation:** Sydney Mulamfu, David Chisompola.

**Formal analysis:** Sydney Mulamfu, David Chisompola.

**Funding acquisition:** Sepiso K. Masenga.

**Investigation:** Sydney Mulamfu, David Chisompola.

**Methodology:** David Chisompola, Sepiso K. Masenga.

**Project administration:** Sepiso K. Masenga.

**Resources:** Sepiso K. Masenga.

**Software:** Sepiso K. Masenga.

**Supervision:** Sepiso K. Masenga.

**Validation:** Sydney Mulamfu, David Chisompola, Martin Chakulya, John Nzobokela, Phinnoty Mwansa, Benson M. Hamooya, Joreen P. Povia, Sepiso K. Masenga.

**Visualization:** Sydney Mulamfu, David Chisompola, Martin Chakulya, John Nzobokela, Phinnoty Mwansa, Benson M. Hamooya, Joreen P. Povia, Sepiso K. Masenga.

**Writing – original draft:** Sydney Mulamfu, David Chisompola, Martin Chakulya, John Nzobokela, Phinnoty Mwansa, Benson M. Hamooya, Joreen P. Povia, Sepiso K. Masenga.

**Writing – review & editing:** Sydney Mulamfu, David Chisompola, Martin Chakulya, John Nzobokela, Phinnoty Mwansa, Benson M. Hamooya, Joreen P. Povia, Sepiso K. Masenga.

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
