## [Decision Letter · Decision Letter 0]

22 Mar 2026

PGPH-D-26-00568

Determinants of Left Ventricular Hypertrophy in a cohort attending a medical clinic: A Comparative Analysis Stratified by Sex

Dear Dr. Chisompola,

Thank you for submitting your manuscript to PLOS Global Public Health. After careful consideration, we feel that it has merit but does not fully meet PLOS Global Public Health’s publication criteria as it currently stands. Therefore, we invite you to submit a revised version of the manuscript that addresses the points raised during the review process.

EDITOR: Dear Author, please make necessary revision based on comments provided by the reviewers.

We look forward to receiving your revised manuscript.

Kind regards,

Zulkarnain Jaafar

Academic Editor

Journal Requirements:

1. Please provide a detailed online Financial Disclosure statement. This is published with the article. It must therefore be completed in full sentences and contain the exact wording you wish to be published.

a) State the initials, alongside each funding source, of each author to receive each grant. For example: “This work was supported by the National Institutes of Health (####### to AM; ###### to CJ) and the National Science Foundation (###### to AM).”

For more information, please go to our submission guidelines:

https://journals.plos.org/globalpublichealth/s/submission-guidelines#loc-financial-disclosure-statement

2. Please ensure that the funders and grant numbers match between the Financial Disclosure field and the Funding Information tab in your submission form. Note that the funders must be provided in the same order in both places as well.

3. Please update your online Competing Interests statement. If you have no competing interests to declare, please state: "The authors have declared that no competing interests exist."

4. Please ensure that you refer to Figure 2 in your text as, if accepted, production will need this reference to link the reader to the figure.

Additional Editor Comments (if provided):

Reviewers' comments:

Reviewer's Responses to Questions

**Comments to the Author**

1. Does this manuscript meet PLOS Global Public Health’s publication criteria? Is the manuscript technically sound, and do the data support the conclusions? The manuscript must describe methodologically and ethically rigorous research with conclusions that are appropriately drawn based on the data presented.

Reviewer #1: Yes

Reviewer #2: Yes

2. Has the statistical analysis been performed appropriately and rigorously?

Reviewer #1: Yes

Reviewer #2: Yes

3. Have the authors made all data underlying the findings in their manuscript fully available (please refer to the Data Availability Statement at the start of the manuscript PDF file)?

Reviewer #1: Yes

Reviewer #2: Yes

4. Is the manuscript presented in an intelligible fashion and written in standard English?

Reviewer #1: Yes

Reviewer #2: Yes

5. Review Comments to the Author

Reviewer #1: Very well presented manuscript regarding the prevalence of left ventricular hypertrophy and its correlation with several risk factors among the two sexes. Given that these data are more or less known in would suggest adding more information regarding the higher prevalence in women in the discussion section as well as strengthening the clinical implications section. Moreover, a central illustration may be added summarising the key findings in a scheme to.make the manuscript more appealing.

Reviewer #2: I would like to thank the authors for submitting this interesting study addressing sex-specific determinants of left ventricular hypertrophy (LVH) in a sub-Saharan African cohort. The topic is clinically relevant, particularly given the growing burden of cardiovascular disease in this population and the importance of sex-based differences in risk stratification.

The study is strengthened by the use of echocardiographic assessment and the attempt to explore sex-stratified determinants, which provides additional insights beyond conventional analyses.

However, several issues should be addressed to improve the scientific rigor and interpretability of the manuscript:

1. Study design and causality

The cross-sectional design limits the ability to infer causal relationships between risk factors and LVH. While this limitation is briefly acknowledged, the discussion should more clearly distinguish association from causation and avoid any causal interpretation.

2. Sample size and model stability in stratified analyses

Although the total sample size is modest (n=333), the number of LVH cases in sex-stratified analyses appears relatively small, particularly in males. This is reflected by wide confidence intervals (e.g., heart failure in males), suggesting potential instability of the regression models. The authors should explicitly report events per variable (EPV) or justify model robustness.

3. Variable selection and potential overfitting

The manuscript mentions multivariable logistic regression, but the process for variable selection is not clearly described. If stepwise selection was used, this should be explicitly stated and justified, as it may increase the risk of overfitting.

4. Definition and measurement of LVH

The criteria used for defining LVH by echocardiography should be clearly specified (e.g., indexed LV mass thresholds, guideline reference), as this is critical for reproducibility and comparison with other studies.

5. Confounding factors

Important confounders such as antihypertensive treatment, duration of hypertension, HIV-related factors (e.g., ART regimen, viral load), and renal function may influence LVH but are not clearly addressed. The authors should clarify whether these were considered or discuss them as limitations.

6. Interpretation of sex differences

The finding that metabolic factors are more prominent in females while hemodynamic factors dominate in males is interesting but should be interpreted cautiously given the limited sample size and cross-sectional design. The discussion should avoid overgeneralization.

7. Clarity and language

The manuscript would benefit from minor language editing for clarity and consistency.

Overall, this is a meaningful study with potential clinical relevance. With the above revisions and clarification of methodological aspects, the manuscript would be significantly strengthened.

6. PLOS authors have the option to publish the peer review history of their article (what does this mean?). If published, this will include your full peer review and any attached files.

**Do you want your identity to be public for this peer review?** For information about this choice, including consent withdrawal, please see our Privacy Policy.

Reviewer #1: **Yes:** Afendoulis Dimitrios

Reviewer #2: **Yes:** Jong-Hwa Ahn, MD, PhD

Figure Resubmissions:

---

## [Decision Letter · Decision Letter 1]

7 Apr 2026

PGPH-D-26-00568R1

Determinants of Left Ventricular Hypertrophy in a cohort attending a medical clinic: A Comparative Analysis Stratified by Sex

Dear Dr. Chisompola,

Thank you for submitting your manuscript to PLOS Global Public Health. After careful consideration, we feel that it has merit but does not fully meet PLOS Global Public Health’s publication criteria as it currently stands. Therefore, we invite you to submit a revised version of the manuscript that addresses the points raised during the review process.

EDITOR: Dear Author, please make necessary revisions as mentioned by the reviewer.

We look forward to receiving your revised manuscript.

Kind regards,

Zulkarnain Jaafar

Academic Editor

Journal Requirements:

Additional Editor Comments (if provided):

Reviewers' comments:

Reviewer's Responses to Questions

**Comments to the Author**

1. If the authors have adequately addressed your comments raised in a previous round of review and you feel that this manuscript is now acceptable for publication, you may indicate that here to bypass the “Comments to the Author” section, enter your conflict of interest statement in the “Confidential to Editor” section, and submit your "Accept" recommendation.

Reviewer #1: All comments have been addressed

Reviewer #2: All comments have been addressed

2. Does this manuscript meet PLOS Global Public Health’s publication criteria? Is the manuscript technically sound, and do the data support the conclusions? The manuscript must describe methodologically and ethically rigorous research with conclusions that are appropriately drawn based on the data presented.

Reviewer #1: Yes

Reviewer #2: Yes

3. Has the statistical analysis been performed appropriately and rigorously?

Reviewer #1: Yes

Reviewer #2: Yes

4. Have the authors made all data underlying the findings in their manuscript fully available (please refer to the Data Availability Statement at the start of the manuscript PDF file)?

Reviewer #1: Yes

Reviewer #2: No

5. Is the manuscript presented in an intelligible fashion and written in standard English?

Reviewer #1: Yes

Reviewer #2: Yes

6. Review Comments to the Author

Reviewer #1: The authors have successfully revised their manuscript according to the reviewers comments. Having the majority of recommendations addressed i find the manuscript fit for publication.

Reviewer #2: Thank you for the revised manuscript. The study addresses a clinically relevant topic, and the revised version is improved. In particular, the authors have clarified the echocardiographic definition of LVH, better described the variable selection strategy for multivariable models, expanded the discussion of sex-specific findings, and more clearly acknowledged the limitations of the cross-sectional design and the limited events-per-variable in the sex-stratified analyses.

The main results appear internally consistent. The overall findings, including the associations of LVH with age, female sex, hypertension, and heart failure, are supported by the data presented. The sex-stratified analyses are also potentially informative, although they should remain interpreted as exploratory given the small number of LVH events in males and the resulting wide confidence intervals.

I have only a few remaining issues before acceptance:

Data availability: The current statement indicates that the data are available only upon request, subject to a data sharing agreement and IRB approval. Please ensure that this fully complies with the journal’s data sharing policy, or provide a more explicit justification for any restriction and clarify whether de-identified underlying data can be deposited or otherwise shared in a policy-compliant manner.

Final proofreading and formatting cleanup: Please carefully remove remaining track-changes/formatting artifacts and merged text fragments in the revised manuscript. A few visible wording and formatting issues remain and should be corrected in the final version.

Reference list check: Please carefully review the references for duplication and consistency. At least one duplicate reference appears to remain in the current version.

Tone of interpretation: The revised discussion is improved, but the sex-specific conclusions should remain cautiously phrased given the cross-sectional design and model instability in subgroup analyses.

Overall, this is a meaningful and improved manuscript, and I believe it would be suitable for publication after these minor final corrections.

7. PLOS authors have the option to publish the peer review history of their article (what does this mean?). If published, this will include your full peer review and any attached files.

**Do you want your identity to be public for this peer review?** For information about this choice, including consent withdrawal, please see our Privacy Policy.

Reviewer #1: **Yes:** Afendoulis Dimitrios

Reviewer #2: No

 Figure Resubmissions:

---

## [Editor Report · Decision Letter 2]

13 Apr 2026

Determinants of Left Ventricular Hypertrophy in a cohort attending a medical clinic: A Comparative Analysis Stratified by Sex

PGPH-D-26-00568R2

Dear Chisompola ,

We are pleased to inform you that your manuscript 'Determinants of Left Ventricular Hypertrophy in a cohort attending a medical clinic: A Comparative Analysis Stratified by Sex' has been provisionally accepted for publication in PLOS Global Public Health.

Best regards,

Zulkarnain Jaafar

Academic Editor